# SPARSE REWARD-ADAPTIVE GENERATIVE FLOW NETWORKS

## ABSTRACT

Generative Flow Networks (GFlowNets) are an emerging class of algorithms for learning policies that sample objects according to an unnormalized reward distribution. While theoretically appealing, in practice, GFlowNets often suffer from training instabilities and mode collapse in environments with sparse rewards. These limit their applicability in a wide range of problems in which high-reward samples are valuable but sparse. In this paper, we identify and analyze three key challenges in training GFlowNets within sparse-reward environments and propose simple and targeted methods to mitigate each of them. Through extensive evaluation across various benchmark environments spanning both discrete and continuous problems, we demonstrate that our methods significantly improve training stability and policy quality, enabling GFlowNets to more reliably discover and exploit high-reward modes in challenging settings.

## 1 INTRODUCTION

Generative Flow Networks (GFlowNets) (Bengio et al., 2021) present a framework to construct objects from a target space $\mathcal{X}$ by learning a generative policy. This policy samples transitions sequentially to reach end states $x \in \mathcal{X}$ that are distributed proportionally to a predefined, unnormalized target distribution termed reward $R(x)$. GFlowNets have demonstrated utility in areas such as biological sequence discovery (Bengio et al., 2021; Jain et al., 2022), drug design (Shen et al., 2024), adversarial prompt generation for large language models (Lee et al., 2025), combinatorial optimization (Zhang et al., 2023b; Kim et al., 2025a), and diffusion (Lahlou et al., 2023a; Zhang et al., 2024; Sendera et al., 2024). Notwithstanding this wide applicability, recent work found that in sparse-reward environments with infrequent non-zero reward signals (i.e., where only a tiny fraction of trajectories lead to end states associated with significantly positive rewards, as formally defined in Section 3), GFlowNets tend to exhibit an undersampling or complete omission of rare, high-reward modes due to the challenges in systematically exploring such sparse rewards.

In general, to improve the training of GFlowNets, several approaches have been proposed that can be broadly grouped into three categories: ($G_1$) **to enhance exploration** for improved discovery of rewarding regions (Bengio et al., 2021; Rector-Brooks et al., 2023; Pan et al., 2023b; Ikram et al., 2025; He et al., 2025; Sendera et al., 2024), ($G_2$) **to improve convergence and credit assignment** for robust and accurate learning (Malkin et al., 2022; Shen et al., 2023; Vemgal et al., 2023; Madan et al., 2023; Jang et al., 2024), and ($G_3$) **to balance exploration and exploitation** to navigate the search space effectively (Pan et al., 2024; Lau et al., 2023; Chen & Mauch, 2024; Kim et al., 2024; 2025b; Lau et al., 2024; Madan et al., 2025). While a subset of methods within these categories, i.e., Pan et al. (2023b); Shen et al. (2023); Madan et al. (2023); Chen & Mauch (2024); Madan et al. (2025), have demonstrated some success in mitigating the undersampling issue in sparse-reward settings, fundamental limitations persist.

In particular, approaches in $G_1$ can facilitate the discovery of high-reward trajectories during training; however, their effectiveness diminishes when the model struggles to fit the observed trajectories, causing the search to become overly local (Rector-Brooks et al., 2023). Similarly, when exploration is insufficient, $G_2$ approaches tend to converge prematurely and fail to fully capture the reward structure (Jang et al., 2024). Lastly, approaches in $G_3$ facilitate mode discovery and training stability; however, they incur substantial computational overhead by specializing the exploration and exploitation tasks to sub-modules, such as pretraining followed by fine-tuning (Pan et al., 2024) or

teacher-student architectures (Kim et al., 2025b). Here, the effectiveness of each sub-module also remains constrained by the inherent limitations of exploration and convergence effectiveness.

These limitations motivate us to answer the following fundamental question: **What are the core challenges of training GFlowNets in sparse-reward environments?** In this paper, we first uncover the principal factors that prevent effective training of GFlowNets and then develop methods to address exploration and model fitting in such environments.

In summary, our contributions are as follows:

- We identify and analyze three core issues in training GFlowNet under sparse-reward environments: (1) the degeneracy of trajectory balance in underexplored regions; (2) the omission of high-reward states due to insufficient exploration; and (3) sampling-induced instability, which results in high gradient variance and unstable updates.

- To mitigate these issues, we propose three methods: (1) an outlier-based filtering mechanism that rectifies misleading loss signals; (2) a decaying-temperature reward-augmentation scheme that enables a smooth transition from initial exploration to later exploitation; and (3) a mixed-priority replay strategy that balances trajectory sampling by jointly considering reward magnitude and training loss.

- We validate the effectiveness of these methods through comprehensive experiments on both discrete (Hypergrid, a molecule generation task named sEH) and continuous (Gaussian Mixture, Pusher) sparse-reward environments, benchmarking against a suite of baselines.

## 2 PRELIMINARIES

GFlowNets (Bengio et al., 2021) are defined by a finite directed acyclic graph $G = (\mathcal{S}, \mathcal{A})$ where the set of nodes forms the state space $\mathcal{S}$ and the set of directed edges forms the action space $\mathcal{A}$. There exists a single state that has no incoming edges, and it is designated as the initial state $s_0 \in \mathcal{S}$. States with no outgoing edges are called terminal (or end) states and their set is denoted by $\mathcal{X}$. States reachable through outgoing edges from a state are called its children, while the sources of incoming edges to a state are called its parents. A trajectory is defined as a sequence of states $\tau = (s_0 \rightarrow \cdots \rightarrow s_n = x)$ where the end state $x \in \mathcal{X}$ and each action (or transition) $s_i \rightarrow s_{i+1} \in \mathcal{A}$. The reward of a trajectory $R(\tau)$ is assigned based on its end state $R(x)$ and $R(x) > 0, \forall x \in \mathcal{X}$. Each trajectory $\tau$ is associated with a nonnegative function called the flow $F : \tau \rightarrow \mathbb{R}_{\geq 0}$. The flow of a state $s$ is defined as $F(s) = \Sigma_{s \in \tau} F(\tau)$ and the flow of an edge $s \rightarrow s'$ as $F(s \rightarrow s') = \Sigma_{s \rightarrow s' \in \tau} F(\tau)$. The objective of GFlowNets is to learn an edge flow such that the flow of each end state $x \in \mathcal{X}$ satisfies $F(x) \propto R(x)$ for all terminal states $x \in \mathcal{X}$.

The edge flow can be modeled by the forward policy $P_F(s'|s) = \frac{F(s \rightarrow s')}{F(s)}$, which defines the transition probability distribution over the children of $s$. Similarly, the backward policy $P_B(s|s') = \frac{F(s \rightarrow s')}{F(s')}$ specifies the probability distribution over the parents of $s'$. The forward (backward) probability of a trajectory $\tau$ is simply the product of forward (backward) probabilities of each edge, i.e., $P_F(\tau) = \prod_{i=0}^{n-1} P_F(s_{i+1}|s_i)$ and $P_B(\tau) = \prod_{i=0}^{n-1} P_B(s_i|s_{i+1})$.

**Trajectory Balance (TB).** TB is computed on complete trajectories. TB has been demonstrated to accelerate training convergence to allow learning policies that generate longer trajectories (Malkin et al., 2022). The GFlowNet training process involves neural network approximation of the forward policy $P_{F_\theta}$, backward policy $P_{B_\theta}$, and the partition function $Z_\theta$, where $\theta$ represents the learnable parameters. For a given trajectory, the TB objective $\mathcal{L}_{\text{TB}}$ is given by

$$\mathcal{L}_{\text{TB}}(\tau) = \left( \log \frac{Z_\theta \cdot P_{F_\theta}(\tau)}{R(s_n) \cdot P_{B_\theta}(\tau)} \right)^2 \tag{1}$$

It is known that if $\mathcal{L}_{\text{TB}}(\tau) = 0 \quad \forall \tau$, the flow $F_\theta$ associated with $P_{F_\theta}$ satisfies the GFlowNet objective (Malkin et al., 2022). Other training criteria are based on similar justifications, as detailed in Bengio et al. (2023), with an extension to the continuous setting provided in Lahlou et al. (2023a).

## 3 THE PROBLEM OF SPARSE REWARDS

**Definition 1 (Sparse-Reward Environment).** We call an environment sparse-reward if the fraction of high-reward trajectories among all possible trajectories is vanishingly small. Formally, let $\mathcal{T}$ be the set of all possible trajectories, and define the set of high-reward trajectories as $\mathcal{H} = \{\tau \in \mathcal{T} \mid R(\tau) > \epsilon\}$ where $\epsilon > 0$ is chosen such that $\sum_{x \in \mathcal{X}} \epsilon \ll \sum_{x \in \mathcal{X}} R(x)$. This ensures that end states with rewards below $\epsilon$ contribute negligibly to the overall learning objective and can be effectively treated as "zero". An environment with sparse rewards shall satisfy $\frac{|\mathcal{H}|}{|\mathcal{T}|} \approx 0$. A similar definition applies for a continuous environment by replacing cardinalities with the Lebesgue measure.

### 3.1 IMPLICATIONS OF SPARSE REWARDS

The above definition implies that, during the early stage of training in sparse-reward environments, the majority of sampled trajectories yield near zero rewards (i.e., $R(\tau) \leq \epsilon$). The constant stream of low rewards can cause the TB loss to drive the partition function estimate $\log Z_\theta$ to a large negative value. Such pessimistic estimates encourage premature convergence with the model overfitting low reward trajectories. More critically, the appearance of an unseen high-reward trajectory under these conditions can trigger exploding gradients, causing severe instability in training.

Another implication is the lack of intermediate reward signals bridging low- and high-reward trajectories, caused by the large magnitude gap between the "zero" reward $\epsilon$ and meaningful high rewards. Without intermediate reward signals, the model becomes highly sensitive to the composition of each training batch. For instance, a batch dominated by high-reward trajectories and one dominated by low-reward trajectories would drive updates to $\log Z_\theta$ in drastically different directions, causing the estimated reward landscape to shift abruptly and ultimately hindering stable convergence.

### 3.2 CHALLENGES OF TRAINING GFLOWNETS IN SPARSE-REWARD ENVIRONMENTS

By formally analyzing the aforementioned implications, we identify three key challenges of GFlowNets training in sparse-reward environments. The first challenge is unique to GFlowNets, distinguishing them from other related approaches such as reinforcement learning (RL). The second and third challenges are shared with RL and other generative models. Throughout this section, we assume that the neural network policies $P_{F_\theta}$ and $P_{B_\theta}$ have full-support over the set of reward-yielding trajectories, and **our analysis is performed at an intermediate stage of training**.

**(C$_1$) Degeneracy of Trajectory Balance in Underexplored Regions.** During training, there may exist underexplored sample spaces with observed trajectories sharing little or no substructure with other trajectories. In such cases, the TB condition may be satisfied without aligning with the GFlowNet objective, leading to misleading loss signals.

**Definition 2 (Underexplored Region).** Let $\mathcal{X}$ be the set of all end states and let $\mathcal{X}_s \subset \mathcal{X}$. Let $\mathcal{T}^{\text{train}}$ be the set of all trajectories observed during training and the subset that ends within $\mathcal{X}_s$ be $\mathcal{T}_s^{\text{train}}$. Let $\mathcal{T}^{\text{unseen}} = \mathcal{T} \setminus \mathcal{T}^{\text{train}}$, the subset that ends within $\mathcal{X}_s$ is named $\mathcal{T}_s^{\text{unseen}}$. We say $\mathcal{X}_s$ is an underexplored region if $\forall \tau^{(1)} = (\cdots \rightarrow s_n^{(1)}) \in \mathcal{T}_s^{\text{train}}, \exists \tau^{(2)} = (\cdots \rightarrow s_n^{(2)}) \in \mathcal{T}_s^{\text{unseen}}$ such that $s_n^{(1)} = s_n^{(2)}$.

This definition characterizes a situation that may arise during the training of GFlowNets, namely when not all trajectories leading to a given terminal state have been observed yet. Further, the definition naturally extends to the continuous setting by replacing discrete states with intervals; in such cases, analysis remains feasible under the assumption that the transition policies are Lipschitz continuous. With the existence of an underexplored region, the following theorem holds.

**Theorem 1 (Degeneracy of Trajectory Balance in Underexplored Regions).** Given an underexplored region $\mathcal{X}_s \subset \mathcal{X}$, a parameter set $\theta^\star$ can minimize the TB loss $\mathcal{L}_{\text{TB}}$ over the training trajectories $\mathcal{T}^{\text{train}}$ while failing to satisfy $F^\star(x) \propto R(x)$ for $x \in \mathcal{X}_s$. This violation of the objective occurs if the learned parameters underestimate the backward policy $P_{B_\theta^\star}$ and the partition function $Z_\theta^\star$ for trajectories associated with $\mathcal{X}_s$, a condition permitted by the absence of training signal in that region.

An illustration of Theorem 1 is shown in Figure 1, and the proof can be found in Appendix A. This issue is exacerbated in sparse-reward environments, where large portions of the state space remain underexplored and high-reward trajectories are initially rare. Consequently, the model tends to

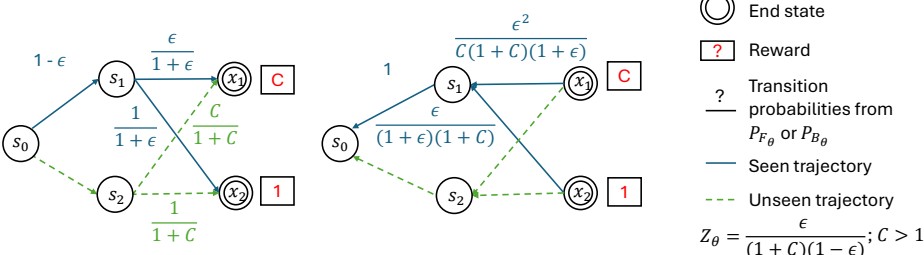

Figure 1: An illustration of a GFlowNet training failure in underexplored regions, which presents two distinct problems. **First**, the model can achieve a zero TB loss with a degenerate solution by assigning an arbitrarily small flow with $0 < \epsilon < C$ to the high-reward state ($x_1$). This is made possible when the partition function $Z_\theta$ and backward policy probabilities $P_{B_\theta}$ are severely underestimated. **Second**, this incorrect status leads to training instability. When the dashed high-reward trajectory is eventually sampled, the TB loss for this trajectory explodes. For $\epsilon \ll 1$, this loss is approximately $\left( \log \frac{\epsilon^2}{(1+C)^2} \right)^2$, which triggers a gradient explosion leading to instability.

underestimate the partition function $Z_\theta$, causing the learned policy to converge prematurely to sub-optimal flows and become trapped in local minima that do not capture the true reward distribution.

**($C_2$) Missed High-reward States.** In sparse-reward environments, trajectories leading to high-reward states represent only a tiny fraction of all possible trajectories. Without a proper exploration strategy and adequate exploration time, one would fail to encounter certain high-reward states during training, causing them to be entirely omitted from the learned GFlowNet policy.

This challenge is simple in concept but hard to fully resolve, especially in large state spaces such as those in the generation of long molecules, a primary application for GFlowNets (Koziarski et al., 2024). In practice, the focus is therefore on whether the training algorithm can effectively discover diverse, high-reward trajectories and their adjacent regions.

**($C_3$) Sampling-Induced Training Instability.** The logarithm operator in the trajectory balance loss function results in numerical instability when processing near-zero values. This results in high gradient variances and even excessively steep gradients, causing unstable parameter updates and suboptimal training outcomes.

This instability is particularly pronounced in two common scenarios driven by the sampling process. First, for the vast majority of trajectories in a sparse-reward setting, the associated reward is near-zero. The model learns to assign these paths near-zero flow values. The logarithm in the loss function is highly sensitive to these near-zero inputs, which causes high loss values and gradient instability. Second, high loss can also arise from high-reward trajectories that are rarely visited by the forward policy, often corresponding to the discovery of previously unexplored high-reward regions.

## 4 METHODS

We devise three targeted and simple methods to mitigate the challenges described in Section 3.2.

### 4.1 BATCH FILTERING (BF)

Theorem 1 suggests that the TB loss can be ineffective for trajectories leading to underexplored regions (e.g., the blue trajectories in Figure 1). To detect such cases, we introduce the statistic:

$$\zeta(\tau) = \log R(\tau) - \log P_{F_\theta}(\tau) - \log P_{B_\theta}(\tau). \tag{2}$$

Intuitively, high-reward trajectories that the model considers improbable (i.e., assigns low forward and backward probabilities) will yield large values of $\zeta(\tau)$. For each training batch **b**, we identify such trajectories as outliers by calculating the mean $\mu_\zeta(\mathbf{b})$ and standard deviation $\sigma_\zeta(\mathbf{b})$ of their $\zeta$ values, we then augment the TB loss as:

$$\mathcal{L}_{\text{TB-BF}}(\tau) = \mathcal{L}_{\text{TB}}(\tau) - \mathbb{I}_{\zeta(\tau) > \mu_\zeta(\mathbf{b}) + c_{\text{upper}}\sigma_\zeta(\mathbf{b})} \big( \log P_{F_\theta}(\tau) + \log P_{B_\theta}(\tau) \big) \tag{3}$$

The consistency of our modified objective with the original GFlowNet goal is guaranteed under the conditions stated in the following theorem.

**Theorem 2 (Existence of a stationary point with batch filtering).** Suppose the backward policy is uniform and satisfies the condition

$$\log P_{B_{\text{uniform}}}(\tau) \geq \mu_{P_{B_{\text{uniform}}}}(\mathbf{b}) - c_{\text{upper}}\sigma_{P_{B_{\text{uniform}}}}(\mathbf{b}), \tag{4}$$

for every $\tau \in \mathbf{b}$ and batch $\mathbf{b} \subseteq \mathcal{T}$, then there exists a solution such that $\mathcal{L}_{\text{TB-BF}}(\tau) = 0 \,\forall \tau \in \mathcal{T}$. Moreover, $\mathcal{L}_{\text{TB-BF}}(\tau) = 0$ for all $\tau \in \mathcal{T}$ yields a policy that samples proportionally to the reward.

The full proof of Theorem 2 can be found in Appendix B. A key observation is that under the TB condition, $\zeta(\tau) = \log Z_\theta - 2\log P_{B_\theta}(\tau)$. The outlier condition on $\zeta(\tau)$ is triggered when the backward probability $\log P_{B_\theta}(\tau)$ is too small. In such cases, the augmented term acts as a corrective mechanism to increase these probabilities. This mechanism can be viewed as a form of soft regularization applied to the backward probability of each trajectory during GFlowNet training.

Based on Theorem 2, $c_{upper}$ should be chosen depending on the problem, or it can be annealed (i.e., gradually increased) during training. In our experiments, we tune a constant $c_{upper}$ for each environment, and the implementation details of BF is provided in Appendix C.1.

## 4.2 Sigmoid Temperature Decay (TD)

To mitigate challenge $C_2$, we introduce a temperature-based reward augmentation mechanism that assigns helper rewards to newly visited states, thereby encouraging the model to explore diverse end states. The temperature parameter gradually decays toward zero so that the augmented reward smoothly converges to the original reward function $R$. This smooth transition is essential: if the reward were to change too abruptly, either in the beginning or near the end of training, it could lead to unstable gradient updates that hinder exploration or degrade performance.

Our design is inspired by techniques in reinforcement learning (Burda et al., 2019a) and simulated annealing (Kirkpatrick et al., 1983), and is theoretically supported by the decaying reward analysis of Pan et al. (2023b). However, unlike previous approaches that multiply the reward or its logarithm by a temperature term (Kim et al., 2024; Zhang et al., 2023a; Hu et al., 2025), we employ an additive temperature formulation. This formulation offers greater flexibility in shaping the search behavior and naturally accommodates the integration of surrogate models to approximate the true reward distribution. In addition, we adopt a sigmoid-shaped decay schedule: the enhanced reward remains stable during the first half of training and then decays almost linearly in log scale. This carefully controlled decay ensures that exploration is encouraged early on, while the training gradually shifts toward exploitation in later stages without destabilizing the trajectory balance loss. Details of the implementation are provided in Appendix C.2.

## 4.3 Mixed Priority (MP) Replay Buffer

To address challenge $C_3$, we introduce a mixed-priority replay mechanism that integrates both reward and loss information. The central idea is that not all trajectories are equally useful for policy learning: trajectories that are highly rewarding but poorly fitted provide particularly informative training signals. By prioritizing such cases, the model is guided to focus on correcting errors in critical regions of the distribution, leading to more stable and effective learning.

Our approach builds on the principle of loss-prioritized replay in reinforcement learning (Schaul et al., 2016) and extends previous work in GFlowNets. In particular, Shen et al. (2023) proposed prioritizing high-reward trajectories to improve sample efficiency and reduce oversampling of low-reward ones. While this improves training performance, it disregards loss values and can destabilize learning when high-reward trajectories are not yet well modeled. Our method addresses this gap by explicitly combining reward and loss in the prioritization scheme.

Formally, the sampling priority for a trajectory $\tau$ is defined as

$$R(\tau)\hat{\mathcal{L}}(\tau) + \frac{1}{|\mathcal{T}_{\text{train}}|} \sum_{\tau \in \mathcal{T}_{\text{train}}} R(\tau)\hat{\mathcal{L}}(\tau) \tag{5}$$

where $\hat{\mathcal{L}}(\tau)$ denotes the estimated relative loss of $\tau$. This formulation encourages sampling of high-loss trajectories while maintaining balanced reward-weighted batches. The mechanism thus continually emphasizes the trajectories most valuable for correcting the model. Implementation details are given in Appendix C.3.

## 5 RELATED WORK

**Advances in GFlowNets Training.** Prior work has addressed each of challenges $C_1$, $C_2$, and $C_3$, yet often in isolation. Our contribution lies in presenting a unified perspective that shows how these challenges collectively impede GFlowNets training in sparse-reward environments, and to help readers contextualize existing enhancements to GFlowNets by clarifying the challenges they address and those that remain unresolved.

For $C_1$, minimizing the trajectory balance loss ensures correct flow when applied to all valid trajectories (Malkin et al., 2022). However, incomplete trajectory coverage can degrade training, as noted by Jang et al. (2024), who proposed updating the backward policy via maximum likelihood. While effective in mitigating degeneracy, this approach can constrain exploration. By contrast, we formalize the notion of underexplored regions and introduce a batch filtering scheme that soft-regularizes backward updates to preserve diversity.

For $C_2$, exploration strategies have been widely studied. Early work introduced $\epsilon$-exploration (Bengio et al., 2021; Malkin et al., 2022), followed by intrinsic rewards (Pan et al., 2023b), uncertainty-driven sampling (Rector-Brooks et al., 2023), domain priors (Ikram et al., 2025), and retrospective augmentation (He et al., 2025). Reward shaping via temperature decay or dynamic scaling has also been explored (Kim et al., 2024; Chen & Mauch, 2024), and hybrid RL–GFlowNet approaches (Lau et al., 2024) bias sampling toward high-reward states. We adopt a complementary approach by adding a sigmoid-decayed auxiliary reward, which is lightweight, broadly applicable, and independent of policy parameterization.

For $C_3$, ordering of sampled trajectories critically affects convergence. Empirical studies highlight issues such as oversampling of low-reward data in replay buffers (Shen et al., 2023) and instability from partition function misestimation (Zhang et al., 2023a). Remedies include reward-prioritized replay (Shen et al., 2023) and adaptive sampling via a teacher policy (Kim et al., 2025b). Related work also explores the balance between exploration and exploitation (Pan et al., 2024; Kim et al., 2024; Madan et al., 2025). Building on these insights, we analyze gradient instability and extend prioritized replay with a reward–loss–mixed scheme.

**Sparse Rewards in RL and Generative Models.** Sparse reward scenarios in reinforcement learning involve infrequent, delayed rewards that impede policy learning due to poor credit assignment and exploration inefficiency, with the latter closely related to $C_2$. Key strategies include: (1) hierarchical decomposition for temporal abstraction (Kulkarni et al., 2016); (2) reward shaping and demonstration-guided learning (Ng et al., 1999; Devidze et al., 2022); (3) experience relabeling via hindsight techniques (Andrychowicz et al., 2017); and (4) intrinsic motivation mechanisms to incentivize exploration (Burda et al., 2019b).

Sparse rewards correspond to low-density regions in generative models. This leads to challenges such as mode collapse (Metz et al., 2017) (similar to $C_1$) and training instability (Salimans et al., 2016) (similar to $C_3$). Song & Ermon (2019) added annealed noise perturbations during training to retain high-dimensional information. Since GFlowNets do not depend on gradients of the target distribution at terminal states, we propose an annealed reward augmentation scheme to encourage sampling from low-reward regions in the early stages of training. While related work has explored decaying exploration in adapting GFlowNet objectives to diffusion tasks (Sendera et al., 2024), our method differs by applying annealing directly to the reward signal.

## 6 EXPERIMENTS

### 6.1 ENVIRONMENTS

We evaluate on four types of environments to verify the effectiveness of our algorithms. Hypergrid and molecule generation (sEH) have discrete state and action spaces, and Gaussian mixture and

Table 1: Performance comparison. Using the trained policy, $100,000$ samples were generated for discrete environments and $10,000$ samples were generated for continuous environments.

| | Hypergrid $D=2$, $H=64$ | | Hypergrid $D=4$, $H=64$ | | sEH | Gaussian Mixture | Multi-objective Pusher |
| --- | --- | --- | --- | --- | --- | --- | --- |
| Success % ($\uparrow$) | # modes ($\uparrow$) | Total $L_1$ ($\downarrow$) | # modes ($\uparrow$) | Total $L_1$ ($\downarrow$) | # modes ($\uparrow$) | # modes ($\uparrow$) | KL Div. ($\downarrow$) |
| TB | $123.2 \pm 30.3$ | $0.33 \pm 0.40$ | $17487.0 \pm 5988.1$ | $0.62 \pm 0.46$ | $25201.4 \pm 687.7$ | $5.50 \pm 0.48$ | $4.0 \pm 6.8$ |
| TB-RP | $114.4 \pm 23.2$ | $0.50 \pm 0.34$ | $7129.4 \pm 4303.2$ | $1.38 \pm 0.31$ | $24676.8 \pm 618.8$ | $5.55 \pm 0.49$ | $2.3 \pm 4.6$ |
| SubTB | $136.4 \pm 8.9$ | $0.17 \pm 0.10$ | $13809.8 \pm 5555.8$ | $0.83 \pm 0.44$ | $24608.8 \pm 1631.7$ | $4.59 \pm 0.52$ | $0.1 \pm 0.1$ |
| GAFN | $102.2 \pm 34.8$ | $0.73 \pm 0.33$ | $679.6 \pm 28.0$ | $1.98 \pm 0.00$ | $7180.8 \pm 6251.9$ | $4.04 \pm 0.24$ | $8.2 \pm 9.9$ |
| PBP-GFN | $84.2 \pm 30.3$ | $0.84 \pm 0.41$ | $3375.8 \pm 914.0$ | $1.68 \pm 0.08$ | $25969.8 \pm 690.2$ | $5.87 \pm 0.30$ | $47.3 \pm 6.9$ |
| Teacher | $\mathbf{144.0 \pm 0.0}$ | $\mathbf{0.09 \pm 0.03}$ | $6201.4 \pm 1183.8$ | $1.46 \pm 0.12$ | $22877.4 \pm 1114.3$ | $4.20 \pm 0.12$ | $6.4 \pm 12.6$ |
| Ours | $\mathbf{144.0 \pm 0.0}$ | $0.11 \pm 0.06$ | $\mathbf{20531.6 \pm 20.2}$ | $\mathbf{0.38 \pm 0.01}$ | $\mathbf{27752.4 \pm 160.2}$ | $\mathbf{3.66 \pm 0.00}$ | $\mathbf{54.6 \pm 4.9}$ |

Multi-objective pusher have continuous state and action spaces. All environments are sparse reward, with detailed specifications and evidence of sparsity provided in Appendix D.

**Hypergrid.** This is a grid-like environment introduced in Bengio et al. (2021) parameterized by side length $H$, dimension $D$, and three reward coefficients $R_0$, $R_1$, and $R_2$. We use a sparser version of the reward function by setting $R_0 = 10^{-10}$ and $R_1 = 0$. We use $D = 2$ and $D = 4$ with $H = 64$ for our experiments. For evaluating performance, we use the empirical $L_1$ error between the sampled distribution and the reward.

**sEH.** This task, introduced in Bengio et al. (2021) and later adopted in several GFlowNet variants (Shen et al., 2023; Madan et al., 2023; Pan et al., 2023a), generates molecules by sequentially attaching blocks to a molecular graph. Molecules are constructed from 18 blocks (2 stems, 6 per molecule), with rewards provided by a proxy scoring model. Following the setup in Shen et al. (2023), the environment includes $34,012,224$ end states. To intensify sparsity, we assign a near-zero reward ($10^{-10}$) to the bottom $99.9\%$ of candidates and define the remaining $0.1\%$ as modes.

**Gaussian Mixture.** We introduce a synthetic continuous environment where state transitions follow a truncated Gaussian distribution with learnable mean and variance. Each episode begins with the center position $(0.5, 0.5)$ and runs up to 10 steps. In our experiments, we set $\delta = 0.02$ and measure performance by the KL divergence between the empirical sample distribution and the ground truth.

**Multi-objective Pusher.** Pusher is a benchmark multi-joint robotic arm environment in Gymnasium (Towers et al., 2024). To better evaluate GFlowNet capabilities, we modify the original single-goal task by introducing three fixed target locations. In each episode, the agent must maneuver the end effector to push the object to any one of these three stationary goals shown in Figure 5(c). Performance is measured as an adjusted success rate, where each target can contribute at most one-third to the total score. Thus, if the agent consistently reaches only one target, the maximum achievable score is capped at $33.33\%$; any higher value implies it has covered additional targets.

## 6.2 BASELINES

We compare our solution methods against several baselines. Here, we focus on trajectory balance (Malkin et al., 2022) (**TB**) and its improvements since they have been shown to outperform detailed balance and flow matching. We compare with subtrajectory balance (Madan et al., 2023) (**SubTB**) since the authors claim it improves GFlowNet training in sparse environments and with trajectory balance using a prioritized replay buffer (Shen et al., 2023) (**TB-RP**) as we propose a modification to their approach for sparse environments. We compare with generative augmented flow networks (Pan et al., 2023b) (**GAFN**), which adds intrinsic rewards to address sparsity. We also consider pessimistic backward policy GFlowNets (Jang et al., 2024) (**PBP-GFN**) since it can reduce the adverse effects of underexplored regions. Finally, we include the adaptive teacher (Kim et al., 2025b) (**Teacher**), which employs a teacher policy to concentrate sampling on regions where the student incurs higher TB losses.

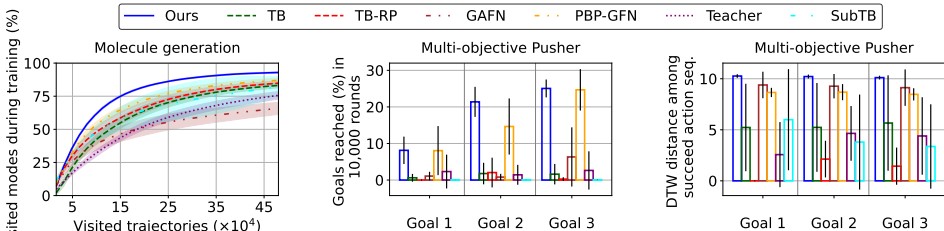

Figure 2: Sample diversity in sEH and Multi-objective Pusher environments. *Left:* Discovered high-reward modes versus the visited trajectories for sEH. The shaded regions show $\pm 1$ standard deviation across five runs. *Middle and Right:* The success rate and diversity scores (higher is better) computed from 10,000 trajectories produced by the trained models for multi-objective pusher.

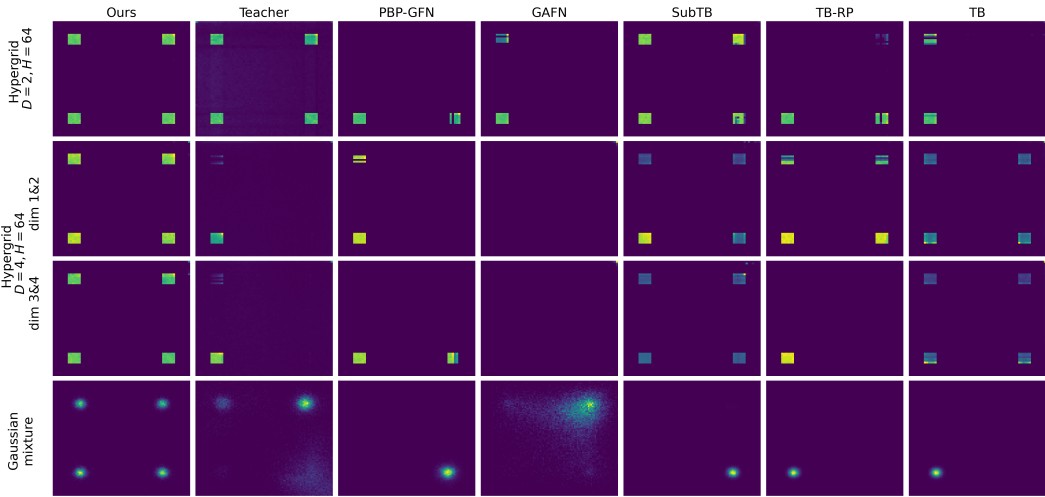

Figure 3: Learned patterns of all methods on the Hypergrid and Gaussian Mixture environments.

## 6.3 Performance Comparison

Table 1 reports the performance of each approach–environment pair using inference data from the trained forward policy. Our methods consistently yield competitive performances across all settings and show the lowest variance over random seeds, demonstrating strong and stable performance.

For the Hypergrid and Gaussian mixture environments, we further visualize the end state distribution and confirm that our approach successfully learns the global reward structure as shown in Figure 3. A visualization of the learned patterns during the training process is provided in Appendix E.1.

For the sEH and multi-objective pusher tasks, we evaluate the diversity of the results since a key objective of GFlowNets is to discover a wide range of high-reward trajectories (Jain et al., 2023). As shown in Figure 2, in sEH, our methods require only 50% of the trajectories compared to the baselines to discover 80% of high-reward modes, demonstrating superior sample efficiency. For the multi-objective pusher task, diversity is quantified using the average Dynamic Time Warping (DTW) distance (Salvador & Chan, 2007) between trajectory pairs that reach the same goal. Our methods not only achieve the highest success rates but also generate the most diverse trajectories.

The suboptimal performance of Teacher in Table 1 could be due to differences between our implementation and the original work. In Kim et al. (2025b), the teacher model employed a uniform backward policy coupled with local search. However, to ensure fair comparison across all evaluated methods, we implemented a trainable backward policy without local search. Under these conditions, we observe that Teacher demonstrates competitive performance in environments with moderately sparse rewards, but performance degrades in highly sparse reward distributions.

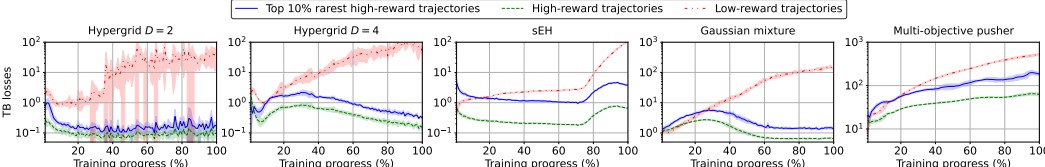

Figure 4: Training losses are reported for different trajectory types. Rare high-reward trajectories are defined as those with reward $> 10^{-3}$ whose $P_{F_\theta}(\tau)$ lies in the bottom $10\%$ of replay buffer samples within each $1\%$ training window.

### 6.4 ANALYSIS OF RESULTS

In this section, we highlight key insights into the role of our proposed mechanisms, including batch filter (BF), temperature decay (TD), and mixed-priority replay (MP), based on the numerical results. Hyperparameter sensitivity analyses for TD and BF and comprehensive ablations of the three mechanisms are provided in Appendices E.2 and E.3.

**BF Promotes Policies for Diverse Trajectory Generation.** As discussed in Section 4.1, BF acts as a soft regularizer on backward probabilities, constraining them away from extremely small values. This encourages the learned policy to generate more diverse trajectories, though it can also interfere with convergence. Our experiments reflect both effects. Figure 2 illustrates BF's influence, and the ablation results in Table 4 (Appendix) further confirm its role: in Pusher, the strong performance is largely attributable to BF, and BF variants in sEH consistently discover more modes than their non-BF counterparts. Nonetheless, in some cases, incorporating BF slightly reduces final performance.

**TD is Effective But Requires Careful Calibration.** TD plays a critical role in the ablation results of Table 4, strongly affecting both mean performance and variance across many environments. To better understand the usability of TD, we perform a sensitivity analysis by varying the initial temperature and decay rate (Appendix E.2). A slower decay rate allows for the initial high temperatures to be maintained longer in the early stages, thereby extending the exploration period. The results indicate that (1) the initial temperature should not be set too high, as it can overwhelm the reward signal, and (2) the optimal decay rate is task-dependent: for discovery tasks such as molecular generation, slower decay is beneficial, while in other tasks, maintaining high temperatures for too long can hinder training the policy to match the true reward distribution.

**MP is More Likely to Sample Rare High-reward Trajectories.** MP prioritizes high-reward trajectories with high training loss. As shown in Figure 4, rare high-reward trajectories are consistently associated with larger loss values, making them more likely to be sampled under MP. At the same time, low-reward trajectories often exhibit even higher loss magnitudes, reinforcing the need for reward-aware prioritization. Figure 4 also highlights that in the multi-objective pusher, policy updates fail to keep pace with the rapid growth of loss. In the multi-objective pusher, policy updates fail to keep pace with the rapid growth of loss, explaining the poor performance of pure TD in the ablation results. This limitation is mitigated by BF and MP, which yield additional performance gains when combined with TD.

## 7 CONCLUSION AND DISCUSSION

We identify three key challenges in training GFlowNets under sparse rewards and introduce targeted mechanisms that improve performance, diversity, and stability. Extensive experiments demonstrate that our sparse-adaptive strategies substantially enhance training across a variety of challenging sparse-reward environments.

While effective, our mechanisms are primarily diagnostic. A critical direction for future research is therefore the development of a theoretically grounded framework that formally addresses these challenges. Such an approach would provide a more principled and robust foundation for training GFlowNets in sparse-reward environments.

## REPRODUCIBILITY STATEMENT

Proofs of theorems in the main paper can be found in Appendices A and B. Pseudocode of our methods and their hyperparameters are given in Appendix C. Environment details and chosen hyperparameters for each baseline can be found in Appendix D. The source code of our methods and implementations of baselines are included as supplementary material.

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

# A  PROOF OF THEOREM 1

To simplify notation in the appendix, for a trajectory or a subtrajectory $\tau = (s_j \to \cdots \to s_k)$, we denote $P_{F_\theta}(\tau) = \prod_{i=j}^{k-1} P_{F_\theta}(s_{i+1}|s_i)$, $P_{B_\theta}(\tau) = \prod_{i=j}^{k-1} P_{B_\theta}(s_i|s_{i+1})$.

**Theorem 1 (Degeneracy of Trajectory Balance in Underexplored Regions).** Given an underexplored region $\mathcal{X}_s \subset \mathcal{X}$, a parameter set $\theta^\star$ can minimize the TB loss $\mathcal{L}_{\text{TB}}$ over the training trajectories $\mathcal{T}^{\text{train}}$ while failing to satisfy $F^\star(x) \propto R(x)$ for $x \in \mathcal{X}_s$. This violation of the objective occurs if the learned parameters underestimate the backward policy $P_{B_\theta^\star}$ and the partition function $Z_\theta^\star$ for trajectories associated with $\mathcal{X}_s$, a condition permitted by the absence of training signal in that region.

**Lemma 1 (Degeneracy of Trajectory Balance in a Special Underexplored Region).** Consider an underexplored region $\mathcal{X}_s^{\text{last}}$ with the property that, for every state $s_n \in \mathcal{X}_s^{\text{last}}$ and every trajectory in $\mathcal{T}_s^{\text{train}}$, there exists another trajectory ending in $\mathcal{X}_s^{\text{last}}$ whose final transition is not included in $\mathcal{T}^{\text{train}}$. Under this condition, a parameter set $\theta^\star$ can minimize the TB loss $\mathcal{L}_{\text{TB}}$ over the training trajectories $\mathcal{T}^{\text{train}}$ while failing to satisfy $F^\star(x) \propto R(x)$ for $x \in \mathcal{X}_s$. This violation of the objective occurs if the learned parameters underestimate the backward policy $P_{B_\theta^\star}$ and the partition function $Z_\theta^\star$.

**Lemma 2 (Composite State Representation).** Any two consecutive state transitions in a trajectory can be equivalently represented as a single composite state. Formally, for any transitions $(s \to s' \to s'')$, define a composite state $\tilde{s} = (s, s')$. Then, the two-step transition can be represented as $\tilde{s} \to s''$. Under this representation, the trajectory balance conditions remain invariant, enabling analysis of GFlowNet training at various granularities.

Theorem 1 is a direct consequence of Lemmas 1 and 2. Lemma 1 establishes that in underexplored regions where the final transitions into certain states are missing from the training data, the trajectory balance (TB) loss signal can be inefficient for updating the policy to match the target distribution. Lemma 2 provides a straightforward observation that any two consecutive transitions can be treated as a composite state, allowing the result of Lemma 1 to generalize to multi-step underexplored regions. Since Lemma 2 is immediate, the proof of Theorem 1 reduces to proving Lemma 1.

*Proof.* Let $\mathcal{X}_s^{\text{last}}$ be an underexplored region defined in Lemma 1, the subset of the training trajectories $\mathcal{T}^{\text{train}}$ that ends within $\mathcal{X}_s^{\text{last}}$ be $\mathcal{T}_s^{\text{train}}$. We begin the proof by constructing a forward transition function $P_{F_\theta^\star}$, backward transition function $P_{B_\theta^\star}$, and partition function $Z_{\theta^\star}$ starting from a state such that the TB objective is satisfied for all $\tau \in \mathcal{T}^{\text{train}} \setminus \mathcal{T}_s^{\text{train}}$. The existence of such a $\theta^\star$ is guaranteed under the assumption that the GFlowNet problem is solvable (i.e., nontrivial).

Then we show that for any trajectory $\tau = (s_0 \to s_1 \to \cdots \to s_{n-1} \to s_n) \in \mathcal{T}_s^{\text{train}}$, arbitrary flow within a non-empty range can be assigned to these trajectories to meet the trajectory balance objective. Furthermore, when $Z_{\theta^\star}$ can fall below a certain threshold, it is possible to assign near-zero flow to high-reward trajectories while allocating the remaining flow to low-reward trajectories.

Since $s_n \in \mathcal{X}_s^{\text{last}}$, there exists another trajectory $\tau' = (s_0 \to s_1' \to \cdots \to s_{n-1}' \to s_n) \in \mathcal{T}_s^{\text{unseen}}$ such that the transition $s_{n-1}' \to s_n \notin \mathcal{T}_s^{\text{train}}$.

The trajectory balance condition for a trajectory $\tau$ is given by

$$Z_{\theta^\star} \prod_{i=0}^{n-1} P_{F_{\theta^\star}}(s_{i+1}|s_i) = R(s_n) \prod_{i=0}^{n-1} P_{B_{\theta^\star}}(s_i|s_{i+1}) \tag{6}$$

Rearranging, we get

$$\prod_{i=0}^{n-1} P_{F_{\theta^\star}}(s_{i+1}|s_i)$$

$$= \frac{\overbrace{P_{B_{\theta^\star}}(s_{n-1}|s_n)R(s_n) \prod_{i=0}^{n-2} P_{B_{\theta^\star}}(s_i|s_{i+1})}^{c_2(\tau)}}{Z_{\theta^*}} \tag{7}$$

Let $\mathcal{S}_{n-1}$ be the set of all states that are the parents of $s_n$. Since $P_{B_{\theta^\star}}$ is a probability measure, $P_{B_{\theta^\star}}(s_{n-1}|s_n)$ satisfies

$$
\begin{aligned}
& P_{B_{\theta^\star}}(s_{n-1}|s_n) \\
& = 1 - \underbrace{\sum_{\tilde{s}_{n-1}\in\mathcal{S}_{n-1}\setminus\{s_{n-1},s'_{n-1}\}} P_{B_{\theta^\star}}(\tilde{s}_{n-1}|s_n)}_{c_1(\tau)} \\
& \quad - P_{B_{\theta^\star}}(s'_{n-1}|s_n)
\end{aligned}
\tag{8}
$$

Given that the transition $s'_{n-1} \to s_n$ is unseen in the training data, it does not participate in calculating the trajectory balance loss. Therefore, $P_{B_{\theta^\star}}(s'_{n-1}|s_n)$ can be an arbitrary value between 0 and $1 - \sum_{\tilde{s}_{n-1}\in\mathcal{S}_{n-1}\setminus\{s_{n-1},s'_{n-1}\}} P_{B_{\theta^\star}}(\tilde{s}_{n-1}|s_n)$.

Let $c_1(\tau) = 1 - \sum_{\tilde{s}_{n-1}\in\mathcal{S}_{n-1}\setminus\{s_{n-1},s'_{n-1}\}} P_{B_{\theta^\star}}(\tilde{s}_{n-1}|s_n)$, we have $P_{B_{\theta^\star}}(s_{n-1}|s_n)$ can be chosen arbitrarily from $(0, c_1(\tau))$. Here, we observe that the backward transition probability $P_{B_{\theta^\star}}(s_{n-1} \mid s_n)$ can be severely underestimated, which in turn drives the forward flow $P_{F_{\theta^\star}}(\tau)$ toward zero, regardless of the reward value $R(s_n)$.

Let $c_2(\tau) = c_1(\tau)R(s_n)\Pi_{i=0}^{n-2}P_{B_{\theta^\star}}(s_i|s_{i+1})$. Then, we have that the righthand side of Equation 7 can be an arbitrary number in $(0, \frac{c_2(\tau)}{Z_{\theta^*}})$. When $Z_{\theta^*} \leq c_2(\tau)$, the transition probability $P_{F_{\theta^\star}}(s_n \mid s_{n-1})$ can be arbitrarily chosen from $(0,1)$ while still satisfying the trajectory balance loss. This demonstrates that $F^*$ can significantly deviate from the GFlowNet objective over some $x \in \mathcal{X}_s^{\text{last}}$. Moreover, when $Z_{\theta^*} = \inf_{\tau\in\mathcal{T}_s^{\text{train}}} c_2(\tau)$, $F^\star$ can satisfy the TB condition over $\mathcal{T}^{\text{train}}$ while arbitrarily poorly meeting the GFlowNet objective of proportionality over $\mathcal{X}_s^{\text{last}}$.

$\square$

## B    PROOF OF THEOREM 2

Theorem 2. Suppose the backward policy is uniform and satisfies the condition

$$
\log P_{B_{\text{uniform}}}(\tau) \geq \mu_{P_{B_{\text{uniform}}}}(\mathbf{b}) - c_{\text{upper}}\sigma_{P_{B_{\text{uniform}}}}(\mathbf{b}),
\tag{9}
$$

for every $\tau \in \mathbf{b}$ and batch $\mathbf{b} \subseteq \mathcal{T}$, then there exists a solution such that $\mathcal{L}_{\text{TB-BF}}(\tau) = 0$ for all $\tau \in \mathcal{T}$. Moreover, $\mathcal{L}_{\text{TB-BF}}(\tau) = 0$ for all $\tau \in \mathcal{T}$ yields a policy that samples proportionally to the reward.

*Proof.* The objective $\mathcal{L}_{\text{TB-BF}}(\tau)$ is defined as:

$$
\mathcal{L}_{\text{TB-BF}}(\tau) = \mathcal{L}_{\text{TB}}(\tau) - \mathbb{I}_{\zeta(\tau)>\mu_\zeta(\mathbf{b})+c_{\text{upper}}\sigma_\zeta(\mathbf{b})}\left(\log P_{F_\theta}(\tau) - \log P_{B_\theta}(\tau)\right)
$$

Since $\mathcal{L}_{\text{TB}}(\tau)$ is a squared term and thus non-negative, a global minimum of zero is possible. A state of $\mathcal{L}_{\text{TB-BF}}(\tau) = 0$ for all $\tau$ requires that two conditions are met simultaneously:

1. $\mathcal{L}_{\text{TB}}(\tau) = 0$ for all $\tau \in \mathcal{T}$.

2. The augmentation term is inactive for all trajectories, i.e., the indicator function $\mathbb{I}_{\zeta(\tau)>\mu(\mathbf{b})+c_{\text{upper}}\sigma(\mathbf{b})}$ is zero for all $\tau$ in any batch $\mathbf{b}$.

Our proof strategy is to demonstrate the existence of at least one policy that satisfies both conditions, thereby showing that the zero-loss state is achievable.

From the foundational theory of GFlowNets (Bengio et al., 2023), for any choice of backward policy $P_B$, there exists a corresponding forward policy $P_F$ and partition function $Z_\theta$ that satisfy the TB condition, making $\mathcal{L}_{\text{TB}}(\tau) = 0$. As per the theorem's premise, let us choose the backward policy to be uniform, $P_B(\tau) = P_{B_{\text{uniform}}}(\tau)$. For this choice, a valid GFlowNet solution exists, satisfying Condition 1.

Now, we must verify that this solution also satisfies Condition 2. When the TB condition is met ($\mathcal{L}_{\text{TB}}(\tau) = 0$), the statistic $\zeta(\tau)$ simplifies to:

$$\zeta(\tau) = \log Z_\theta - 2 \log P_{B_\theta}(\tau) \tag{10}$$

For our chosen solution with a uniform backward policy, this becomes:

$$\zeta(\tau) = \log Z_\theta - 2 \log P_{B_{\text{uniform}}}(\tau) \tag{11}$$

The triggering condition of the augmentation term can be rewritten as

$$\log Z_\theta - 2 \log P_{B_{\text{uniform}}}(\tau) > \log Z_\theta - 2\mu_{P_{B_{\text{uniform}}}}(\mathbf{b}) + 2c_{upper}\sigma_{P_{B_{\text{uniform}}}}(\mathbf{b}) \tag{12}$$

Further simplify and get

$$\log P_{B_{\text{uniform}}}(\tau) < \mu_{P_{B_{\text{uniform}}}}(\mathbf{b}) - c_{upper}\sigma_{P_{B_{\text{uniform}}}}(\mathbf{b})$$

which is what we assumed to be always false. $\qquad\square$

## C   METHOD DETAILS

### C.1   DETAILS OF BATCH FILTERING

Based on Theorem 1, we use batch-level statistics of the signal $\zeta(\tau) = \log R(\tau) - \log P_{F_\theta}(\tau) - \log P_{B_\theta}(\tau)$ to identify potentially overfit trajectories. We compute the batch-level mean and standard deviation of $\zeta$ and check if each trajectory $\tau$ is an outlier with $\zeta(\tau) > \mu + c_{upper}\sigma$. The pseudocode is given in Algorithm 1.

---
**Algorithm 1** Batch filtering

---
**Require:** Upper threshold $c_{upper}$.
  Sample one batch of trajectory $\mathbf{b} = \{\tau_1, \tau_2, \cdots, \tau_n\}$ from the replay buffer.
  Collect the reward $R(\tau_k)$, compute the forward transition $\log P_{F_\theta}(\tau_k)$, backward transition $\log P_{B_\theta}(\tau_k)$ for each trajectory $\tau_k$, $k = 1, \cdots, n$.
  For each trajectory $\tau \in \mathcal{T}_{\text{batch}}$, calculate the signal $\zeta(\tau) = \log R(\tau) - \log P_{F_\theta}(\tau) - \log P_{B_\theta}(\tau)$.
  Calculate the mean and standard deviation of $\zeta$ to obtain $\mu(\mathbf{b})$, $\sigma(\mathbf{b})$ respectively.
  **for all** $\tau \in \mathbf{b}$ **do**
    **if** $\zeta(\tau) > \mu(\mathbf{b}) + c_{upper}\sigma(\mathbf{b})$ **then**
      Set the loss of $\tau$ according to Equation (3).
    **end if**
  **end for**

---

### C.2   DETAILS OF TEMPERATURE DECAY

We augment the reward function $R$ with an additive augmented reward $R'$ to encourage exploration. By decaying these added values over time, the learning objective gradually aligns with the original problem.

A linear reward decay would result in an exponential increase in GFlowNet loss during later training stages due to the $\log R$ term in the loss function. Hence, we devise the sigmoid decay that encourages exploration during the early stages of training while reserving sufficient training iterations for the model to adapt to the gradually diminishing augmented rewards. Algorithm 2 presents the pseudocode.

In the experiments, we use a constant function $R'(\tau) = 1$.

### C.3   DETAILS OF MIXED PRIORITY REPLAY BUFFER

We use $R(\tau)\hat{\mathcal{L}}(\tau)$ as the weight to sample from the replay buffer, where $\hat{\mathcal{L}}$ represents the relative loss of the trajectory $\tau$, normalized to a value in $[0.1, 10.0]$.

---

**Algorithm 2** Sigmoid Temperature Decay

---

**Require:** Total steps $T$, initial temperature $\alpha_0$, decay rate $\beta_0$, augmented reward function $R'$.
   **for** $t = 1$ to $T$ **do**
      Sample one batch of trajectories $\{\tau_1, \tau_2, \cdots, \tau_n\}$ from the replay buffer.
      Perform GFlowNet updates by treating the reward of $\tau_k$ as $R(\tau_k) + \alpha R'(\tau_k)$.
      Update the temperature: $\alpha = \frac{\alpha_0}{1 + 10^{10 - \beta_0(1 - t/T)}}$.
   **end for**

---

To ensure that low-reward trajectories are not omitted, we offset the weights by adding the mean weight, balancing the sampling probability between low-reward and high-reward trajectories when the replay buffer contains mostly low reward data. Additionally, to prevent the sampler from disproportionately selecting low-reward trajectories due to small losses for high-reward trajectories, we introduce a recovery step. This step restores the loss of high-reward trajectories to 1 if they have not been visited for an extended period. The pseudocode is shown in Algorithm 3.

---

**Algorithm 3** Prioritized experience replay buffer with mixed weighting.

---

   **procedure** WEIGHTEDSAMPLING(Batch size $n$)
      Sample $n$ records by the weights $\left(R(\tau)\hat{\mathcal{L}}(\tau) + \frac{1}{|\mathcal{T}_{\text{train}}|} \sum_{\tau' \in \mathcal{T}_{\text{train}}} R(\tau')\hat{\mathcal{L}}(\tau')\right)$
      Return the index of samples $\{i_{\tau_1}, i_{\tau_2}, \cdots, i_{\tau_n}\}$, the sampled trajectories $\{\tau_1, \tau_2, \cdots, \tau_n\}$.
   **end procedure**
   **procedure** UPDATEWEIGHT(Indexes $\{i_{\tau_1}, i_{\tau_2}, \cdots, i_{\tau_b}\}$, Losses $\{l_{\tau_1}, l_{\tau_2}, \cdots, l_{\tau_b}\}$)
      **for all** $\tau \in \mathcal{T}_{\text{train}}$ **do**
         **if** $R(\tau) > 90$ percentile of the observed $R$ in $\mathcal{T}_{train}$ **then**
            Update $\hat{\mathcal{L}}(\tau) \leftarrow 0.95\hat{\mathcal{L}}(\tau) + 0.05$
         **end if**
      **end for**
      **for all** $i_\tau \in \{i_{\tau_1}, i_{\tau_2}, \cdots, i_{\tau_n}\}$ **do**
         Update $\hat{\mathcal{L}}(\tau) \leftarrow \frac{9.9 l_\tau}{\max(\{l_{\tau_j}\}_{j=1,\cdots,n}) - \min(\{l_{\tau_j}\}_{j=1,\cdots,n})} + 0.1$
      **end for**
   **end procedure**

---

# D EXPERIMENTAL DETAILS

## D.1 WHY ARE THESE ENVIRONMENTS SPARSE?

We introduce two methods to evaluate the condition described in Section 3. The first is empirical. A typical untrained policy, which is uniform in discrete settings or Gaussian or uniformly distributed over a bounded range in continuous spaces, is used to sample trajectories. If the proportion of nonzero-reward trajectories is close to zero, the environment is considered sparse. The second method applies to problems defined by a size parameter $n$, such as the number of bits or spatial dimensions. In these cases, the reward is considered sparse if the fraction of high-reward outcomes vanishes as $n$ becomes sufficiently large.

To measure the sparsity of rewards in our environments, we apply the first approach mentioned above. Specifically, we use a uniform random sampler to generate $10^5$ trajectories for a discrete action space. For the Gaussian mixture and pusher, we sample trajectories from a truncated normal distribution with a mean of 0 and standard deviations of 0.05 and 0.5, respectively. We then calculate the percentage of high-reward trajectories, with the results presented in Table 2. All experiments use a random seed of 42.

## D.2 IMPLEMENTATION

We strictly follow Lahlou et al. (2023b) to implement the GFlowNet training algorithm with TB and SubTB losses, with modifications to add our approaches and adapt to the standard APIs provided by Gymnasium (Towers et al., 2024). To ensure correctness, we test its performance in the hypergrid

| | Hypergrid | | sEH | Gaussian Mixture | Pusher |
| --- | --- | --- | --- | --- | --- |
| | $D = 2$ | $D = 4$ | | | |
| | 0.140 | 0.022 | 0.099 | 0.827 | 0.242 |

Table 2: Sparsity of environments: percentage of high-reward trajectories $(R(\tau) > 1e - 3)$ in $100,000$ attempts with untrained policies.

example $8 \times 8 \times 8 \times 8$ using the same hyperparameters specified in Lahlou et al. (2023b). Our results confirm that the implementation achieves performance consistent with the original.

For GAFN, we follow the source code provided by Pan et al. (2023b). For PBP-GFN, we build on the implementation described in Jang et al. (2024). Since the original code only supports discrete environments, we extend it to continuous settings by maximizing the log-likelihood of the backward transition probabilities of the observed trajectories. For Teacher-Student training (Teacher), we follow the code provided in Kim et al. (2025b).

### D.3 OFF-POLICY TRAINING

As our approach follows an off-policy training paradigm, we set the replay buffer size to 100,000, the training frequency to 16, and perform 10 gradient updates per training step using the Adam optimizer. For each experiment, we initially collect 96 trajectories using an untrained policy, as we find this improves early training stability. All experiments are repeated five times with random seeds $470, 3825, 4444, 8888, 9528$. These training configurations are applied across all our experiments.

To ensure a fair comparison under off-policy training, we apply reward-prioritized training (RP) to all baselines except vanilla TB, following the insights from Vemgal et al. (2023), which show that RP improves mode discoverability and accelerates training. This choice mitigates the risk of performance degradation due to a biased replay buffer and ensures that observed differences are due to the baselines' core algorithms rather than incompetent sampling strategies.

For PBP-GFN, we follow the off-policy training hyperparameters, maximizing the probability of the probabilities of backward transition in sampled batches of observed trajectories $8$ times before each training step.

Teacher–Student training was originally proposed for the on-policy learning paradigm. However, we note it can also be applied off-policy. As suggested in Vemgal et al. (2023), on-policy learning is generally more challenging; to ensure fairness in our comparison, we combine it with reward-prioritized replay (RP). Following the ablation results in Kim et al. (2025b), we use $\alpha = 0$. As local search is not always available, the behavioral policy is limited to the teacher and the student, with equal probability assigned to each when sampling new trajectories.

### D.4 COMPUTATIONAL RESOURCES

All experiments were conducted on an internal compute cluster managed by SLURM, with jobs scheduled using reproducible SLURM job scripts. Each compute node was equipped with a single NVIDIA A10 GPU (24GB VRAM), and each job requested 64GB of RAM and 16 CPU cores.

Training times varied by environment: approximately 2 hours for the Gaussian mixture environment and up to 4 hours for the 4D Hypergrid. SubTB incurred significantly higher computational costs on Hypergrid tasks, with some instances requiring over 24 hours to complete.

### D.5 HYPERGRID

**Reward Function.** The reward function of hypergrid in (Bengio et al., 2021) is defined using three coefficients $R_0$, $R_1$, and $R_2$. Given height $H$, dimension $D$, and an end state $x$ with coordinates $(x^1, x^2, \cdots, x^D)$, the reward function is expressed as:

$$R(x) = R_0 + R_1 \prod_{d=1}^{D} \mathbb{I}\left[|\frac{x^d}{H-1}| \in (0.25, 0.5)\right] + R_2 \prod_{d=1}^{D} \mathbb{I}\left[|\frac{x^d}{H-1} - 0.5| \in (0.3, 0.4)\right]$$

| Environment | Method | Learning Rate | Batch Size | Activation Function | Network Structure | Epsilon Random |
|---|---|---|---|---|---|---|
| Hypergrid $D=2, H=64$ | TB TB-RP GAFN PBP-GFN SubTB Teacher | $5e-3$ | 16 | LeakyReLU | [256, 256] | Off |
| Hypergrid $D=4, H=64$ | TB TB-RP GAFN PBP-GFN SubTB Teacher | $1e-4$ | 16 | LeakyReLU | [256, 256] | Off Off On Off Off Off |
| sEH | TB TB-RP GAFN PBP-GFN SubTB Teacher | $1e-3$ $1e-3$ $1e-4$ $1e-3$ $1e-3$ $1e-3$ | 16 | LeakyReLU | [256, 256] | On |
| Gaussian Mixture | TB TB-RP GAFN PBP-GFN SubTB Teacher | $1e-3$ $1e-3$ $1e-4$ $5e-3$ $1e-3$ $1e-3$ | 16 16 32 16 16 32 | LeakyReLU | [256, 256] | Off Off On On Off On |
| Multi-objective Pusher | TB TB-RP GAFN PBP-GFN SubTB Teacher | $1e-5$ $1e-5$ $1e-5$ $1e-4$ $1e-5$ $1e-5$ | 32 32 32 16 32 32 | LeakyReLU | [128, 128] [128, 128] [256, 256] [128, 128] [128, 128] [256, 256] | Off |

Table 3: Hyperparameters chosen for each environment and method.

For our experiments, we set $R_0 = 10^{-10}$, $R_1 = 0$ and $R_2 = 2$ to make the reward function sparser than previously studied. The reward fnction with these configurations is shown in Figure 5(a). Under these settings, in the $64 \times 64$ environment, only 4% of the end states yield rewards greater than the base reward ($10^{-10}$). In the more challenging $64 \times 64 \times 64 \times 64$ variant, only $0.16\%$ of the end states have high rewards.

**States and Action.** We follow Bengio et al. (2021) and use the one-hot encoding for each dimension of the coordinates, resulting in an input state of dimension $D \times H$. The actions also follow Bengio et al. (2021), consisting of one exit action and operations of incrementing the selected coordinate by 1 at each step.

**Hyperparameters.** We follow Malkin et al. (2022) and adopt a multilayer perceptron (MLP) architecture with two hidden layers of 256 units each. The learning rate for $Z_\theta$ is set to be 100 times that of the forward and backward policies. We perform a grid search to tune the hyperparameters, selecting the learning rate from $\{0.0001, 0.0005, 0.001, 0.005\}$, the activation function from ReLU and LeakyReLU, and choosing whether to apply $\epsilon$-random exploration (starting at 0.1 and decaying linearly during training). The selected hyperparameters are summarized in Table 3.

For both hypergrid environments, we tuned the hyperparameters of our proposed methods starting from the optimal configuration used for TB-RP. For the temperature decay mechanism (TD), we evaluated the initial temperature $\alpha_0$ of the set $\{1.0, 0.1, 0.01, 0.001\}$ and the decay rate $\beta_0$ of

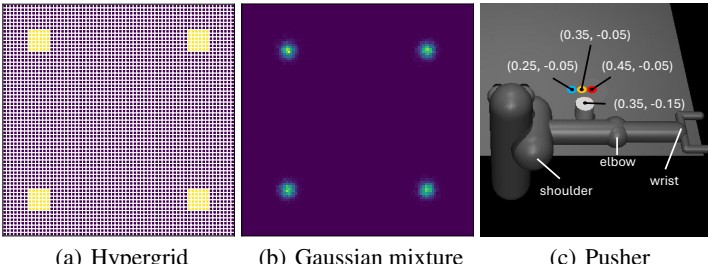

| (a) Hypergrid | (b) Gaussian mixture | (c) Pusher |

Figure 5: (a) and (b) show the reward function of the hypergrid and Gaussian mixture environments, respectively. (c) shows the multi-objective pusher environment with the three targets.

$[10, 40]$. We selected $\alpha_0 = 0.001$ and $\beta_0 = 10$. For batch filtering, we tested the upper threshold $c_{\text{upper}}$ from $\{1, 2, 3\}$ and selected $c_{\text{upper}} = 3$.

Since LeakyReLU consistently outperforms ReLU across experiments, we adopt LeakyReLU as the activation function for all subsequent experiments.

### D.6 sEH

**Reward Function.** We use the proxy model from (Shen et al., 2023) to compute rewards, which estimates the binding energy of a molecule to the soluble epoxide hydrolase (sEH) protein target. Following (Shen et al., 2023), we set the reward exponent to 6. To introduce sparsity into the environment, we assign a near-zero reward value of $10^{-10}$ to the bottom 99.9% of proxy-evaluated states.

**States and Actions.** We follow (Shen et al., 2023) to use 18 blocks with 2 stems, and use 6 blocks per molecule. These result in state dimension of $18 \times 6 = 108$ and the action dimension of $2 \times 18 + 1 = 37$.

**Hyperparameters.** We adopt a multilayer perceptron (MLP) architecture. The learning rate for $Z_\theta$ is set to be 100 times that of the forward and backward policies. We perform a grid search to tune the hyperparameters, selecting the policy learning rate from $\{0.0001, 0.0005, 0.001, 0.005\}$, the neuron size between $[128, 128]$ and $[256, 256]$, and choosing whether to apply $\epsilon$-random exploration (starting at 0.1 and decaying linearly during training). The selected hyperparameters are summarized in Table 3.

We tuned the hyperparameters of our proposed methods starting from the optimal configuration used for TB-RP. For the temperature decay (TD) mechanism, we evaluated the initial temperature $\alpha_0$ of the set $\{1.0, 0.1, 0.01, 0.001\}$ and the decay rate $\beta_0$ of $[10, 40]$. We selected $\alpha_0 = 0.001$ and $\beta_0 = 40$. For batch filtering, we tested the upper threshold $c_{\text{upper}}$ from $\{1, 2, 3\}$ and selected $c_{\text{upper}} = 1$.

### D.7 GAUSSIAN MIXTURE

**Reward Density.** This problem is continuous, and the reward density function is modeled as a mixture of four independent Gaussian distributions as shown in Figure 5(b). The mean values are $\mu_1 = (0.2, 0.2), \mu_2 = (0.8, 0.2), \mu_3 = (0.2, 0.8), \mu_4 = (0.8, 0.8)$, with each distribution having a standard deviation of 0.02. Given an end state $x \in [0, 1] \times [0, 1]$, the reward density $R(x)$ is defined as

$$R(x) = \frac{1}{4} \sum_{i=1}^{4} \frac{1}{0.02\sqrt{2\pi}} \exp\left(-\frac{\|x - \mu_i\|^2}{2(0.02)^2}\right) + 10^{-10}$$

**States and Actions.** We encode the state to represent the temporal information (current step in a trajectory length of 10 as the maximum trajectory length is 11) and the relative quadrant compared to the initial state. Specifically, values greater than 0.5 and less than 0.5 are encoded into two distinct dimensions, resulting in an input state vector of length 40.

We sample actions from a truncated Gaussian distribution, with the mean ranging from $(-0.1, 0.1)$, the standard deviation from $(0.001, 0.1)$, and the range limited to $(-0.2, 0.2)$. We also add a dimension to the action to indicate the exit of the episode.

**Hyperparameters.** We use a multilayer perceptron (MLP) architecture. The learning rate for $Z_\theta$ is set to be 100 times that of the forward and backward policies. We perform a grid search to tune the hyperparameters, selecting the policy learning rate from $\{0.0001, 0.0005, 0.001, 0.005\}$, the neuron size between $[128, 128]$ and $[256, 256]$, the batch size between 16 and 32, and choosing whether to apply $\epsilon$-random exploration (starting at 0.1 and decaying linearly during training). The selected hyperparameters are summarized in Table 3.

We tuned the hyperparameters of our proposed methods starting from the optimal configuration used for TB-RP. For the temperature decay (TD) mechanism, we evaluated the initial temperature $\alpha_0$ of the set $\{1.0, 0.1, 0.01, 0.001\}$ and the decay rate $\beta_0$ of $[10, 40]$, we selected $\alpha_0 = 1.0$ and $\beta_0 = 10$. For batch filtering, we tested the upper threshold $c_{\text{upper}}$ from $\{1, 2, 3\}$ and selected $c_{\text{upper}} = 2$.

**Approximating KL-Divergence.** Given $N$ points sampled from the GFlowNet model, we approximate the KL-divergence between the learned GFlowNet sampler and the target distribution $P$ as follows:

1. We fit a kernel density estimator (KDE) $Q$ on the $N$ samples.
2. We create a mesh grid over the $1 \times 1$ area with 10000 points, with step size 0.01 on either dimension.
3. For each grid point $x$, we compute the value $P(x) \log \frac{P(X)}{Q(x)}$
4. We approximate the KL-divergence as the KL-divergence as the mean of these computed values over all grid points.

### D.8 MULTI-OBJECTIVE PUSHER

This is an environment to exemplify the potential limitation of applying our approach to sparse-reward environments with huge state spaces.

**Reward Density.** This environment is based on "Pusher" implemented in Gymnasium Towers et al. (2024), a sparse-reward environment with 7 degrees of freedom in the reinforcement learning community Zhang et al. (2021). The original objective is to control a robotic arm to push a cylinder towards a goal position. We modified the reward function by adding a second goal position to meet our purpose of GFlowNet to discover novel robotic control strategies that achieve specific conditions. Given the current coordinates $x = (x^{(1)}, x^{(2)})$ of the cylinder centroid, the goal coordinates $\mu_1 = (0.45, -0.05)$ and $\mu_2 = (0.25, -0.05)$, $\mu_3 = (0.35, -0.05)$, the reward density is defined as

$$R(x) = \max(50000(\mathbb{I}(\|x - \mu_1\| < 0.05) + \mathbb{I}(\|x - \mu_2\| < 0.05) + \mathbb{I}(\|x - \mu_3\| < 0.05), 10^{-10})$$

**States and Actions.** We use the states provided by the original "Pusher" environment, excluding the goal position, resulting in an 19-dimensional state representation. Additionally, we encode the time step as a 9-dimensional vector (the maximum trajectory length is 10), producing a final input state with a length of 28.

Unlike the original environment, we fix the initial state: both the initial position and velocity of the pusher arm are set to constant values of 0, and the initial position of the cylindrical object is set to $(-0.1, -0.1)$. We also modify the control frequency by setting the step size to 0.5 seconds and restricting each episode to at most 10 time steps.

The action space is 4-dimensional, derived from the original control inputs by excluding the last three dimensions corresponding to wrist rotation, which are fixed to zero.

**Hyperparameters.** We use a multilayer perceptron (MLP) architecture. The learning rate for $Z_\theta$ is set to be 100 times that of the forward and backward policies. We perform a grid search to tune the hyperparameters, selecting the policy learning rate from $\{0.00001, 0.00005, 0.0001, 0.0005, 0.001, 0.005\}$, the neuron size between $[128, 128]$ and $[256, 256]$, the batch size between 16 and 32, and choosing whether to apply $\epsilon$-random exploration (starting at 0.1 and decaying linearly during training). The selected hyperparameters are summarized in Table 3.

We tuned the hyperparameters of our proposed methods starting from the optimal configuration used for TB-RP. For the temperature decay (TD) mechanism, we evaluated the initial temperature $\alpha_0$ of the set $\{1.0, 0.1, 0.01, 0.001\}$ and the decay rate $\beta_0$ of $[10, 40]$. We selected $\alpha_0 = 1.0$ and $\beta_0 = 10$. For batch filtering, we tested the upper threshold $c_{\text{upper}}$ from $\{0, 0.5, 1, 2\}$. We found that setting $c_{\text{upper}} = 0$ yielded the highest successful rate, suggesting all end states are underexplored for this environment.

**Dynamic Time Warping (DTW) Distance.** For each trained model, we generate trajectories and identify those that successfully reach a specific goal. For each pair of successful trajectories corresponding to the same goal, we compute the Dynamic Time Warping (DTW) distance between their action sequences, as defined in Equation 13:

$$\text{DTW}(a_i, a_j) = \min_{\pi} \sum_{(t,t') \in \pi} \|a_i^t - a_j^{t'}\|_2,$$

(13)

where $a_i$ and $a_j$ denote the action sequences of two successful trajectories, and $\pi$ represents a warping path aligning their time steps. Here, we obtain $\text{DTW}(a_i, a_j)$ using the Python `fastdtw` package.

For each goal, we then average the pairwise DTW distances over all successful trajectory pairs. We use the resulting mean DTW value to quantify the diversity of successful control strategies and report it in the rightmost panel of Figure 2.

# E  ADDITIONAL EXPERIMENTAL RESULTS

## E.1  LEARNED PATTERNS DURING TRAINING

Figure 6 illustrates the distributions learned by our approaches during training with random seed 470. The results show effective exploration, demonstrating both effective exploration and reliable convergence to high-reward states.

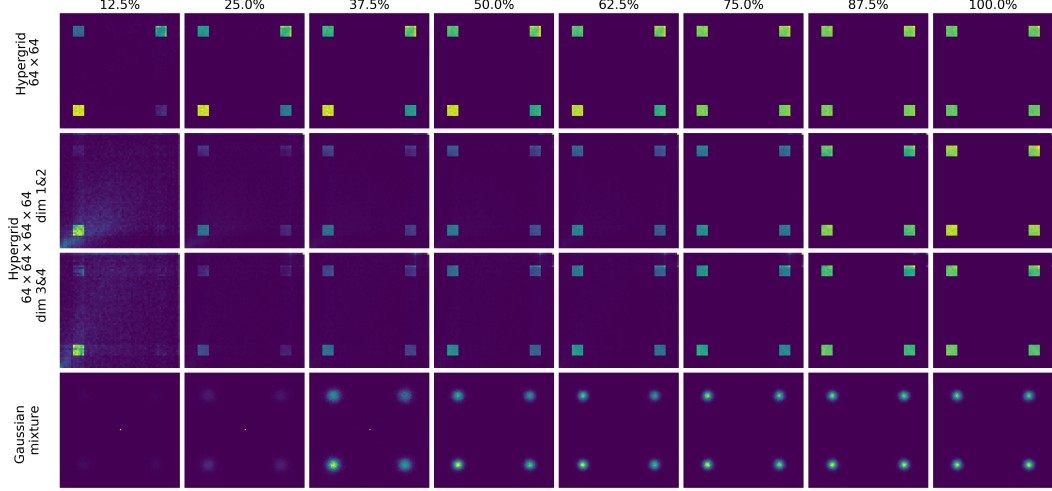

Figure 6: Learned patterns of our approaches for Hypergrid and Gaussian mixture environments.

## E.2  SENSITIVITY ANALYSIS

Figure 7 presents the sensitivity analysis of our method with respect to key hyperparameters. Across all tested environments, our approach performs robustly over a broad range of hyperparameter values, supporting their practical utility since extensive tuning is not required.

| | Hypergrid (L$_1$ Error ↓) | | sEH | Gaussian Mixture | Multi-objective Pusher |
|---|---|---|---|---|---|
| | $D = 2$ | $D = 4$ | (# modes ↑) | (KL Div. ↓) | (Success Rate ↑) |
| BF + MP + TD | **0.037 ± 0.002** | **0.283 ± 0.004** | **30730.6 ± 172.6** | 3.191 ± 0.001 | **0.541 ± 0.032** |
| BF + RP + TD | 0.046 ± 0.005 | 0.312 ± 0.018 | 29030.8 ± 28.0 | **3.190 ± 0.001** | 0.321 ± 0.093 |
| BF + MP | 0.792 ± 0.371 | 1.461 ± 0.259 | 29449.2 ± 106.8 | 5.586 ± 0.605 | 0.305 ± 0.103 |
| BF + RP | 0.863 ± 0.202 | 1.534 ± 0.067 | 28038.4 ± 135.6 | 5.312 ± 1.083 | 0.212 ± 0.024 |
| MP + TD | 0.038 ± 0.002 | 0.284 ± 0.008 | 29910.0 ± 140.7 | 3.211 ± 0.018 | 0.094 ± 0.186 |
| RP + TD | 0.040 ± 0.004 | 0.303 ± 0.009 | 28685.8 ± 204.3 | 3.213 ± 0.031 | 0.000 ± 0.000 |
| BF + TD | 0.046 ± 0.004 | 0.923 ± 0.143 | 29070.8 ± 279.2 | 3.188 ± 0.001 | 0.035 ± 0.007 |
| MP | 0.841 ± 0.335 | 1.205 ± 0.327 | 29154.4 ± 349.5 | 6.715 ± 0.851 | 0.197 ± 0.215 |
| RP | 0.620 ± 0.327 | 1.483 ± 0.175 | 27461.0 ± 723.6 | 6.198 ± 0.997 | 0.028 ± 0.056 |
| BF | 0.724 ± 0.292 | 1.154 ± 0.410 | 28236.8 ± 330.9 | 4.172 ± 1.172 | 0.134 ± 0.120 |
| TD | 0.041 ± 0.003 | 0.988 ± 0.132 | 28593.4 ± 70.4 | 3.211 ± 0.018 | 0.031 ± 0.017 |
| Vanilla | 0.520 ± 0.432 | 0.786 ± 0.443 | 27139.2 ± 707.8 | 5.498 ± 1.039 | 0.029 ± 0.036 |

Table 4: Ablation study results. Each cell reports the mean and standard deviation across 5 seeds.

Several insights emerge from this analysis. First, increasing the level of exploration (higher values of temperature) generally improves training stability, as indicated by the standard deviations. This effect is especially noticeable with higher temperature decay rates, where the temperature drops more rapidly in the later stages of training. Such schedules encourage more aggressive exploration, leading to more stable outcomes.

Second, while setting the initial temperature too low can cause insufficient exploration, we observe that overly large initial temperatures can also lead to incorrect flow within, as evident in the Hypergrid environments. To avoid this, the initial temperature should be set two to three orders of magnitude smaller than the high-reward values, but can be much closer to the scale of the total rewards. For instance, we observe that an initial temperature of 0.01 performs well in the 4D Hypergrid, while a higher value, like 1.0, is suitable for the Gaussian mixture environment. We note that adding these minimal rewards significantly alters the original problem.

Finally, batch filtering proves critical in the multi-objective Pusher environment. Performance improves when more trajectories are identified as overfitted and treated by our interventions. This highlights the importance of incorporating batch filtering in sparse-reward problems with large state spaces, where most sampled trajectories tend to fall into underexplored regions.

### E.3 ABLATION STUDY

To assess the contribution of each component to overall performance, we conduct an ablation study across all environments. Table 4 shows the results.

The combination of batch filtering (BF), mixed-priority replay buffer (MP), and temperature decay (TD) consistently yields the best results, highlighting the complementary benefits of these components.

We observe that in certain environments, partial combinations of our proposed components can already yield competitive results. For instance, applying TD alone performs well in the 2D Hypergrid and Gaussian mixture environments, while combining MP and TD effectively addresses the more complex 4D Hypergrid. This aligns with our understanding that sparse-reward challenges involve different aspects, and their difficulties vary across the tested environments.

For environments such as molecule generation and multi-objective pusher, which involve significantly larger and more complex state spaces, the full combination of BF, MP, and TD leads to the best performance. We note that these environments are particularly relevant to real-world applications, where the search space is vast and achieving both high sample efficiency and diverse high-reward trajectory discovery is of paramount interest.

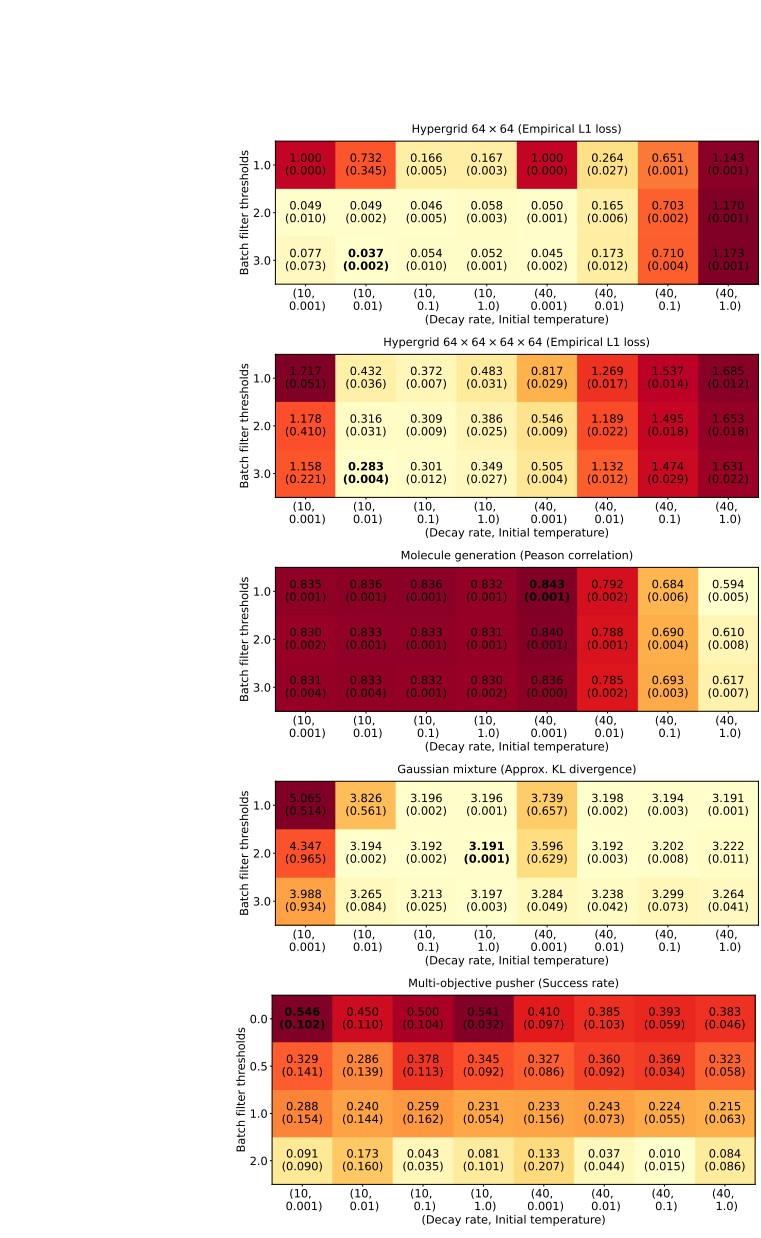

Figure 7: Sensitivity results. The cell value (top) shows the average performance metric for the environment, and in brackets shows the standard deviation of the performance metric across five seeds.

