# OpenReview forum: "Sparse Reward-Adaptive Generative Flow Networks"
_ICLR.cc/2026/Conference — ICLR 2026 Conference Withdrawn Submission_

### Official Review · Reviewer_CSMp · 2025-10-22

**Soundness:** 2
**Presentation:** 3
**Contribution:** 2
**Rating:** 2
**Confidence:** 4

**Summary:**

This paper addresses the challenge of training GFlowNets in sparse-reward environments, where only a small fraction of the space contains rewards separated from zero. The authors identify three key issues: degeneracy of trajectory balance in underexplored regions, missed high-reward states, and sampling-induced training instability. To address them, a GFlowNet training algorithm is proposed, combining the techniques of 1) batch filtering that identifies overfit trajectories, 2) temperature decay to promote exploration, 3) mixed-priority replay buffers to stabilize the training. The effectiveness of the proposed approach is showed in an experimental evaluation on 4 environments, and the effect of individual techniques is examined in an ablation study.

**Strengths:**

I found the writing clear and easy to follow. The studied problem has high practical importance and significance in my opinion. The authors present a number of interesting insights on the challenges of training GFlowNets in sparse-reward environments, and construct an approach that tackles each of them. The presented experimental results show the benefits of the proposed approach in 4 environments in comparison to a number of baselines, additionally highlighting the importance of the aforementioned challenges.

**Weaknesses:**

The main weaknesses of the paper in my opinion are limited novelty of the proposed techniques, as well as toy nature of the most tasks used for experiments.

Temperature decay has already been utilized and studied in GFlowNet literature [1, 2, 3]. Though the authors claim that they modify the decay scheme, the technique itself is already widely known. In addition, the modification should be directly ablated in comparison to the previously employed schemes to show that training actually benefits from it. I did not find such an ablation in the paper.

Prioritized replay buffers have already been utilized for GFlowNet training in the previous literature [4, 5, 6]. While the authors acknowledge the existence of previous works that do prioritization by the reward, they attribute using the loss in the prioritization for GFlowNet training as their novelty. However, [6] already used loss-prioritized replay buffers for GFlowNet training, and this work is not cited in the paper. I suggest that the approach from [6] should be added as a baseline, and an ablation study should be carried out to show that the training actually benefits from the proposed mixed prioritization scheme in comparison to prioritizing solely by the rewards and prioritizing solely by the loss.

In addition, since one of the goals of the proposed methodology is to reduce gradient variances (line 194), I would suggest discussing previous works on variance reduction in GFlowNets [7] in the paper.

Next, I do not fully understand the motivation behind the statistic used for batch filtering (Equation 2). Consider the followign example. Suppose that $Z = 1$ (rewards are normalized), and trajectory balance is satisfied: $\log P_F(\tau) = \log R(x) + \log P_B(\tau)$, meaning that the GFlowNet already performs perfectly for a given trajectory. This also means that $\log R(x) + \log P_B(\tau) - \log P_F(\tau) = 0$. However, the value of the utilized statistic $\zeta(\tau) = \log R(x) - \log P_B(\tau) - \log P_F(\tau)$ can be arbitrary large in this case, and increasing the value of $\log P_B(\tau) + \log P_F(\tau)$ for this trajectory (as done in the proposed modified loss) will only break the trajectory balance. Can you please elaborate on this?

I am also struggling to understand the significance of Theorem 1. In complex GFlowNet environments studied in practice, almost any region will be underexplored during training by the presented definition, and it seems trivial that the perfect sampling policy will not be exactly obtained in this case. This will happen because there are usually too many possible trajectories leading to each terminal state, meaning that some of them will always stay unseen. Even in the very small 2D 64 x 64 hypergrid, he number of trajectories leading to the upper right corner is $C(128, 64) \approx 2 \times 10^{37}$, which will never be explored in its entirety in practice. In my opinion, a more meaningful theoretical result might be to show in some way how exactly the amount of explored trajectories in the region affects the distribution approximation error.

Finally, the set of the environments used for the experimental evaluation is somewhat limited in my opinion. Hypergrids and gaussian mixtures are toy synthetic environments. While sEH is a standard task in the GFlowNet literature, the paper uses a small version of it, containing only 34 million terminal states. This environment can also be considered toy since the space of objects we work with is small enough that it is possible to iterate over all objects in reasonable time, thus the task of training a GFlowNet policy for sampling is artificial and redundant. To demonstrate the practical utility of the proposed method, evaluation on larger environments is crucial, thus I would suggest considering other environments from the GFlowNet literature, e.g. the larger original version of sEH environment from [8] or biological sequence environments widely studied in GFlowNet literature (e.g. AMP and GFP, see [9]).


References:\
[1] Zhang et al. Let the Flows Tell: Solving Graph Combinatorial Optimization Problems with GFlowNets. NeurIPS 2023\
[2] Jang et al. Pessimistic Backward Policy for GFlowNets. NeurIPS 2024\
[3] Kim et al. Learning to Scale Logits for Temperature-Conditional GFlowNets. ICML 2024\
[4] Shen et al. Towards Understanding and Improving GFlowNet Training. ICML 2023\
[5] Vemgal et al. An Empirical Study of the Effectiveness of Using a Replay Buffer on Mode Discovery in GFlowNets. ICML 2023 Workshop\
[6] Tiapkin et al. Generative Flow Networks as Entropy-Regularized RL. AISTATS 2024\
[7] Silva et al. On Divergence Measures for Training GFlowNets. NeurIPS 2024\
[8] Bengio et al. Flow Network based Generative Models for Non-Iterative Diverse Candidate Generation. NeurIPS 2021\
[9] Madan et al. Learning GFlowNets from partial episodes for improved convergence and stability. ICML 2023

**Questions:**

0) See Weaknesses.

1) Is there an error in Table 1 in the column for Pusher environment? It suggests that the metric is KL (which should be minimized), while the largest value is highlited in the table.

2) Pusher is a task that is not commonly studied in GFlowNet literature. Can you please elaborate on it, if the task is to reach specific target locations, why should GFlowNets be applied, not classical RL? Is the task to reach all 3 target locations in one trajectory, or to reach each of the target locations with equal probabilities?

---

### Official Review · Reviewer_Qu9e · 2025-10-27

**Soundness:** 2
**Presentation:** 3
**Contribution:** 1
**Rating:** 2
**Confidence:** 4

**Summary:**

This paper addresses the problem of training Generative Flow Networks (GFlowNets) in sparse-reward environments, where only a small fraction of trajectories yield non-zero rewards. The authors identify three key challenges:

1. Trajectory balance objective can yield a suboptimal solution, given that it was optimized over a subset of trajectories set $\mathcal{T}$. This is typical in sparse reward scenarios;

2. Lack of exploration - inability to uncover all high-reward states;

3. Sampling-induced instability of the training objective.

Authors propose three corresponding mitigation strategies:

- Batch Filtering (BF) – detects and regularizes misleading loss signals using outlier statistics on trajectory likelihoods;
- Sigmoid Temperature Decay (TD) – introduces a smooth reward augmentation that gradually decays, improving early exploration without destabilizing training;
- Mixed-Priority Replay (MP) – combines reward-based and loss-based prioritization in the replay buffer to emphasize informative trajectories.

They evaluate these on both discrete (Hypergrid, sEH molecule generation) and continuous (Gaussian Mixture, Multi-objective Pusher) environmetns.

**Strengths:**

The proposed methods are intuitive and shown to improve performance across both discrete and continuous environments. At the same time, there are significant concerns about the novelty of suggested approaches.

**Weaknesses:**

1. First, all three key issues raised as the main points of the paper, are well-known in the GFlowNet (and RL) literature, as well as methods to mitigate them. Issues with trajectory balance objectives gave rise to various methods, among which the most popular one is SubTB. The usage of prioretized experience replay is also somehow standard, e.g. [Vemgal et al, 2023].

2. It is unclear, why theorem 2 is useful. In Theorem $1$ it was shown, that the trajectory balance objective might yield spurious optimums, given a training subset of trajectories. In Theorem 2, the authors show that if one can ensure some condition (eq. (4)) for all trajectories in the set $\mathcal{T}$, the corresponding GFlowNet induces a proper distribution, but in this setting the TB objective is also satisfied (as shown in the appendix). It would be much more interesting to understand, what happens if this condition is satisfied only for a subset of trajectories $\mathcal{T}_s$, which is somehow "representative enough".

3. Experimental evaluation is significantly limited. Authors evaluate algorithms only on two discrete and two continuous environments, among them the Gaussian mixtures and Hypergrid are rather toyish ones. It is clear that the $2$-dimensional Gaussian mixture example can be solved with literally any reasonable MCMC or diffusion sampler baseline (see e.g. [Vargas et al, 2023]). The sEH example is also rather simple as it has rather small state space.


Minor points:

- As a consequence of weakness $2$ (and generally), it is always controversial to write about "extensive evaluation", as in the abstract.
- There are some issues with reported metrics. In particular, for the Gaussian Mixture task, the smaller number is highlighted in bald (Table 1). I also did not understand, why larger values of KL divergence are highlighted in bald in the same table 1, speaking about the Pusher environment
- Definition 1 is not really a definition, as it is quite unformal.

***References***

[Vemgal et al, 2023] Vemgal, N., Lau, E., Precup, D. (2023). An empirical study of the effectiveness of using a replay buffer on mode discovery in gflownets. arXiv preprint arXiv:2307.07674.

[Vargas et al, 2023] Vargas, F., Grathwohl, W.,Doucet, A. (2023). Denoising diffusion samplers. arXiv preprint arXiv:2302.13834.

**Questions:**

- Do the authors compare their method with TB objective with different exploration strategies?

- Is it possible to provide additional numerical evaluation on more challenging environments with larger state spaces?

---

### Official Review · Reviewer_gwWj · 2025-10-29

**Soundness:** 4
**Presentation:** 2
**Contribution:** 3
**Rating:** 4
**Confidence:** 4

**Summary:**

SImilar to RL ,GFlowNets often suffer from training instabilities and mode collapse in environments with sparse rewards. The author analyze the reason why this happened in GFlowNets, proposed a simple solution and conduct experiment to understand the effects

**Strengths:**

The manuscript aims at solving one of the most foundamental problem shared by GFN ,RL and many other methods: learning frrom sparse reward. The author gave a clear definition of sparse reward. The authors gave a thoughtful explanation of why sparse reward lead to problem in trajectory balance of GFN such as large negative of normalization factor z and instability in training. The author identified 3 reason why this lead to bad GFN optimization: 1)  Degeneracy of Trajectory Balance in Underexplored Regions, 2) Missed High-reward States and 2) Sampling-Induced Training Instability.

The author proposed a combination of 3 methods to mitigate the problem 1) filter out the outlier 2) temperature decay 3) mix replay buffer

Among which the most novel contribution is 1) use a z-score style threshold to exclude large value trajectory .

The authors conducted a series of 5 expeirments , 3 on traditional GFN tasks , one on newly proposed Gaussian continuous task and one on robotic task

**Weaknesses:**

1. This works only focus on trajectory balance loss of GFN, which is closely related to RL methods. There are other GFN loss such as FM, which are not analyzed

2. The sparse reward problem is shared by most RL and other method, not sure if the contribution is unique

3. The 3 approaches to improve are quite incremental instead of foundamental

4. the presentation of the paper( eg. visualization of experimental results) can be improved

**Questions:**

1) to  mitigate degeneracy of Trajectory Balance in Underexplored Regions, the authors uses a simple (mu + std) to calculate threshold, why is this chosen? other options may be better? An ablation study is needed and a theoretical analysis should be provided

2) The author claimd that the the degeneracy of Trajectory Balance of GFN comes from underestimate the backward policy and the partition function Z. Exact behavior on z and P_b need to be analyzed empriclaly

---

### Official Review · Reviewer_8F33 · 2025-11-02

**Soundness:** 3
**Presentation:** 3
**Contribution:** 2
**Rating:** 2
**Confidence:** 5

**Summary:**

The paper studies why GFlowNets struggle with sparse rewards and proposes three simple add-ons:  batch filtering (BF) that down-weights likely degenerate TB signals using a ζ-statistic, a sigmoid temperature-decayed additive reward to reshape the original reward function, and a mixed-priority (MP) replay that prioritizes trajectories by reward times loss. Experiment on On Hypergrid (2D/4D), sEH molecule generation, Gaussian Mixture, and a multi-objective Pusher are conducted to show the effectivnees of proposed add-ones againt a suite of baselines.

**Strengths:**

*  The paper clearly isolate and formalizes three sparse-reward issues—TB degeneracy in underexplored regions, missed high-reward states, and sampling-induced instability.

* BF/TD/MP are easy to implement and generally applicable.

* Reproducibility details and code are included in appendices and supplement.

**Weaknesses:**

* The first issues, "Degeneracy of Trajectory Balance in Underexplored Region" is only specific to TB objective. The other objectives (DB, Sub-TB), which define state flow function $F(s)$ for $s\in \mathcal{S}$  that rather than solely the partition function  $Z=F(s_0)$,are not affected by this problem—or are much less susceptible to it.
   * Since multiple trajectories can terminate at the same terminal state and exhibit the degeneracy noted by the authors, the core idea of GFlowNet learning is to enforces state-level flow balance over a DAG[1], where for any state $s\in \mathcal{S}$, the total incoming flow from its parent states $Pa(s)$ should equals the total outgoing flow from $s$ to its child states $Ch(s)$.  These state-level modeling will compensate the unobserved transitions.

* Theorem 1 is theoretically flawed.  The proof  regarding “trajectory degeneracy”  is built upon the assumption that certain transition probabilities $P(s_{n-1}'|s_n)$ can vary continuously from $0$ to some value $c$. This assumption is generally invalid:
   *  Under a fixed backward policy, those transition probabilities are already determined and cannot vary freely.
   *  For a non-fixed backward policy (usually parameterized by a neural network), the smoothness (or sharpness) of the output distribution is limited, meaning that certain transition probabilities cannot freely change without affecting the others due to normalization constraints.

* For the other two issues identified by the authors, there is no substantially new contribution. As acknowledged by the authors themselves, the proposed variants are largely derived from or adapted from previous approaches. Although the forms may differ, the underlying ideas remain quite similar, rendering the overall contribution incremental rather than conceptually novel.

  * Besides, regarding the third issue, I disagree with the statement that “the logarithm in the loss function is highly sensitive to near-zero inputs, leading to high loss values and gradient instability.” As far as I know, it is standard practice to clamp the minimum value of any logarithmic term to a reasonable negative threshold (e.g., −10), which effectively resolves this problem[2].

* All the considered discrete spaces are relatively small, and therefore do not make the issues caused by sparse rewards particularly severe.

* The experimental results are not reported in a convincing way. Please provide the performance metric values throughout the training process, similar to how the training loss is reported.  Besides, both the mode number and mode diversity matter.


[1] Bengio, Yoshua, et al. "Gflownet foundations." Journal of Machine Learning Research 24.210 (2023): 1-55.

[2] Lahlou, Salem, et al. "torchgfn: A pytorch gflownet library." arXiv preprint arXiv:2305.14594 (2023).

**Questions:**

* What's the exact meaning of $P_{B_\theta}$ or $Z_\theta$ is underestimated ?   Any formal definitions or intuitive explanations?

* What types of trajectory replay strategies and reward temperature settings are used for the baselines (e.g., TB and Sub-TB)?
How does the performance of TB and Sub-TB equipped with prior plug-in methods differ from that when they are equipped with the proposed methods? I am concerned that the comparison may not be fair if the authors simply compare plain versions of TB and Sub-TB against TB equipped with the proposed plug-in methods.

---

### Note · Authors · 2025-11-22

**Comment:**

We sincerely appreciate the reviewers’ time and thoughtful feedback. We are committed to substantially revising the work to address the provided feedback and intend to submit a significantly improved version to a future CS venue.

**Withdrawal Confirmation:**

I have read and agree with the venue's withdrawal policy on behalf of myself and my co-authors.